# Postnatal Maturation of the Blood–Brain Barrier in Senescence-Accelerated OXYS Rats, Which Are Prone to an Alzheimer’s Disease-like Pathology

**DOI:** 10.3390/ijms242115649

**Published:** 2023-10-27

**Authors:** Ekaterina Rudnitskaya, Tatiana Kozlova, Alena Burnyasheva, Daniil Peunov, Michail Tyumentsev, Natalia Stefanova, Nataliya Kolosova

**Affiliations:** Institute of Cytology and Genetics, Siberian Branch of Russian Academy of Sciences (ICG SB RAS), 10 Lavrentyeva Ave., 630090 Novosibirsk, Russia; kozlova@bionet.nsc.ru (T.K.); burnyasheva@bionet.nsc.ru (A.B.); peunovda@bionet.nsc.ru (D.P.); landselur@bionet.nsc.ru (M.T.); stefanovan@bionet.nsc.ru (N.S.); kolosova@bionet.nsc.ru (N.K.)

**Keywords:** Alzheimer’s disease, blood–brain barrier, postnatal development, OXYS rats

## Abstract

Alzheimer’s disease (AD) is an old-age neurodegenerative disorder; however, AD predisposition may arise early in life. Vascular dysfunction makes a big contribution to AD development. Nonetheless, the possible role of early-life vascular dysfunction in AD development is still poorly investigated. Here, using OXYS rats as a suitable model of the most common (sporadic) type of AD, we investigated maturation of the blood–brain barrier (BBB) in the hippocampus and frontal cortex in the first 3 weeks of life. Using RNA-Seq data, we found an altered expression of BBB-associated genes in the middle of the first and second weeks of life in OXYS rats compared to control rats (Wistar strain). Moreover, by immunohistochemistry and electronic microscopy, we revealed a delay of vascularization and of subsequent pericyte coating of blood vessels in OXYS rats. These specific features were accompanied by an accelerated decrease in BBB permeability estimated using Evans blue dye. Notably, almost all of the observed differences from Wistar rats disappeared on postnatal day 20. Nonetheless, the observed features, which are characteristic of the postnatal period, may have long-term consequences and contribute to neurovascular dysfunction observed in OXYS rats late in life, thereby promoting early development of AD signs.

## 1. Introduction

Alzheimer’s disease (AD) is the most common form of senile dementia and is characterized by progressive memory loss against a background of an accumulation of β-amyloid and hyperphosphorilation of tau protein resulting in synaptic dysfunction and death of neurons [1].

Recent studies involving state-of-the-art diagnostic techniques have confirmed that the preclinical period of the sporadic type (>95% of cases [2]) of AD can last for decades [3], but the question of when exactly disease development begins and what contributes to it remains open. There is a growing body of evidence from epidemiological and experimental studies that the predisposition to accelerated aging (which is the main risk factor for AD) can emerge as early as during the early postnatal period, at the age of the completion of brain maturation [4,5,6]. One such risk factor is the formation of aberrant neural circuits under the influence of genetic and/or environmental factors [7]. It was reported that prenatal hypoxia [8] and low weight at birth [9] may be among these factors determining the brain development trajectory as well as the risk of AD.

The results of our studies on senescence-accelerated OXYS rats—a unique model of sporadic AD [10]—have confirmed the validity of this assumption. OXYS rats spontaneously develop all key signs of AD (such as behavioral alterations and learning and memory deficits) simultaneously with the hyperphosphorylation of the tau protein in the hippocampus and cortex, impaired long-term potentiation, and first signs of neurodegeneration at about 3–5 months. With age, neurodegenerative changes in the brains of OXYS rats become amplified against the background of overproduction of amyloid precursor protein and accumulation of β-amyloid, and by the age of 16–18 months reach the well-pronounced stages of AD-like pathology [11]. It is assumed that mitochondrial dysfunction mediates or possibly even initiates pathological molecular cascades of AD-like pathology in OXYS rats [12].

As we have shown previously, accelerated aging of the brain and the development of AD signs in OXYS rats can be attributed to delays in the completion of postnatal brain maturation. We have identified the features of brain maturation in OXYS rats in the early postnatal period (at ages “postnatal day 0” [PND0] to PND20) that may be prerequisites for the development of initial neurodegenerative changes at a later age. It has been found that the completion of brain maturation in OXYS rats occurs against the background of a decrease in astrocytic and microglial support taking place in the hippocampus and prefrontal cortex [13]. The glial support is a key regulator of neural-circuitry functioning [14]. The lack of glial support may be the reason for the decrease in the efficiency of the formation of synaptic contacts revealed in OXYS rats [11], and this notion suggests that this lack is a key event leading to further development of AD signs.

It is well known that astroglia play a key role in the formation of the blood–brain barrier (BBB). The latter is a structure that separates nervous tissue from the blood. The BBB consists of endothelial cells together with a basement membrane, pericytes, and end-foot processes of astrocytes [15]. Endothelial cells of the central nervous system (CNS) are connected to each other via tight junctions (TJs), which limit paracellular diffusion of solutes and ions across the BBB. Endothelial cells express a variety of transporters to provide the CNS with nutrients and specific molecules from the blood as well as to implement the efflux of metabolites and potential toxins from the CNS. These specific properties of CNS endothelial cells are not intrinsic but rather are caused by the brain microenvironment [16]. The permeability of the BBB is actively regulated by different components of a neurovascular unit: endothelial cells, pericytes, smooth muscle cells, astrocytes, microglia, and neurons [17]. The formation of brain vessels in mice starts on embryonic day 10: angioblasts of the perineural vascular plexus invade the neural tube [18]. Nevertheless, these angioblasts give rise to “leaky” blood vessels, and barrier formation starts around embryonic day 15 in mice [17]. The BBB continues to mature postnatally, and cortical capillaries finally acquire their adult morphology by PND20 in rodents [19].

The breakdown of the BBB was recently shown to be a possible mechanism underlying age-related cognitive decline [20]. Neuroimaging studies have revealed early BBB dysfunction in AD and other neurodegenerative disorders. An increase in BBB permeability together with a decrease in the level of the main amyloid-β brain-to-blood transporter—low-density lipoprotein receptor-related protein 1 (LRP1)—accompany the earliest AD stages prior to cognitive dysfunction [21,22]. Nonetheless, the possible link between alterations of BBB formation early in life and the development of neurodegenerative changes remains elusive.

The main aim of this study was to determine the efficiency of BBB formation in the hippocampus and prefrontal cortex of OXYS rats during the period of completion of postnatal brain development and to evaluate their possible contribution to the development of AD-like pathology in these animals.

## 2. Results

### 2.1. Differential Expression of Genes Associated with the BBB in the Hippocampus and Frontal Cortex of OXYS and Wistar Rats

We used the transcriptomic data obtained by Daneman and colleagues [16] as the basis of our examination. Daneman and colleagues reported a list of BBB-enriched transcripts compared to the endothelium of other tissues (liver and lung).

We analyzed the expression of BBB-enriched transcripts in the hippocampus and frontal cortex of OXYS and Wistar rats at PND3 and PND10 (Appendix A). First, we found the number of the BBB gene that changed their expression from PND3 to PND10 both in the hippocampus (187 of 412 genes in OXYS rats; 194 of 412 in Wistar rats) and in the frontal cortex (200 of 412 genes in OXYS rats; 198 of 412 in Wistar rats). The vast majority of these genes changed their expression in the same direction when OXYS and Wistar rats were compared. The only exception was the *Tgfa* gene, whose expression in the frontal cortex increased in Wistar rats and decreased in OXYS rats from PND3 to PND10, resulting in a lower level of *Tgfa* mRNA in the frontal cortex of OXYS rats at PND10, as compared to Wistar rats.

We noticed that in the hippocampus at PND3, the expression of 6% of BBB-enriched transcripts (24 of 412; six differentially expressed genes [DEGs] were upregulated, and 18 DEGs were downregulated) changed in OXYS rats, whereas at PND10, almost 15% of BBB-enriched transcripts were differentially expressed (59 of 412; 25 DEGs were upregulated, and 34 DEGs were downregulated). Among them, at both ages, four DEGs were found to be overexpressed (*Slc7a5*, *Mllt11*, *Egfl8*, and *Alas2*), and 12 DEGs were underexpressed (*Vamp4*, *Sparcl1*, *Serinc5*, *Reck*, *Ptn*, *Mpzl1*, *Lynx1*, *Glul*, *Chn2*, *Cd34*, *Apod*, and *Ak3*). The construction of gene networks revealed that at both ages DEGs formed networks with significant protein–protein interaction (PPI) enrichment (*p* < 0.03 for PND3 and *p* < 0.01 for PND10; Figure 1). Functional enrichment analysis showed that only at PND10 was there one signaling pathway from the KEGG database that was enriched by DEGs: the phenylalanine metabolism pathway (False Discovery Rate [FRD] = 0.04).

In the frontal cortex, we observed that at PND3 almost 15% of BBB-enriched transcripts showed changes in their expression in OXYS rats (58 of 412; 19 DEGs were upregulated, and 39 DEGs were downregulated), whereas at PND10, there were 10% of BBB-enriched differentially expressed transcripts (43 of 412; 15 DEGs were upregulated, and 28 DEGs were downregulated). Among them, only three DEGs proved to be overexpressed (*Tgfb2*, *Marveld2*, and *Lrp8*), and 11 DEGs were underexpressed (*Slco2b1*, *Slc7a1*, *Reck*, *Myo1e*, *Maob*, *Lynx1*, *Lrig1*, *Glul*, *Cd34*, *Apod*, and *Ak3*); meanwhile, the mRNA level of the *Acsl6* gene was high at PND3 and low at PND10, whereas the mRNA level of *Jag2* was diminished at PND3 and elevated at PND10. In the frontal cortex, DEGs formed networks with significant PPI enrichment at both ages (*p* < 0.003 for PND3 and *p* < 0.005 for PND10; Figure 2). According to functional enrichment analysis, the ribosome pathway (KEGG database) was significantly enriched by DEGs (FDR = 0.003).

Additionally, at PND3, the expression of 10 BBB-enriched transcripts proved to be decreased both in the hippocampus and frontal cortex (*Slc38a3*, *Serinc5*, *Reck*, *Mpzl1*, *Lynx1*, *Glul*, *Eps8l2*, *Cd34*, *Apod*, and *Ak3*); at PND10, the expression of six DEGs was excessive (*Tgfb2*, *Slc25a32*, *Jag2*, *Hbaa1*, *Egfl8*, and *Alas2*), and the expression of 19 DEGs was low (*Tfrc*, *Slco2b1*, *Slco1c1*, *Slc1a1*, *Serpine2*, *Rgs5*, *Reck*, *Ptn*, *Maob*, *Lynx1*, *Lrrc49*, *Lrig1*, *Glul*, *Ddc*, *Cdkn2b*, *Cd34*, *Apod*, *Ak3*, and *Acsl6*) in both brain structures.

### 2.2. Differential Expression of Genes Involved in Junction Formation in the Hippocampus and Frontal Cortex of OXYS and Wistar Rats

To estimate the permeability of the BBB, we analyzed the expression of the genes belonging to the “tight junction,” “adherens junction,” and “gap junction” signaling pathways according to the Kyoto Encyclopedia of Genes and Genomes (KEGG) database.

#### 2.2.1. TJs

Closest to the apical membrane, TJ proteins limit paracellular diffusion of solutes and ions across the BBB [23].

Analysis of the transcriptomic data indicated that 105 of 167 genes of the TJ signaling pathway (according to the KEGG Pathway database) changed their expression from PND3 to PND10 in the hippocampus of both Wistar and OXYS rats (Appendix A). The majority of these DEGs were common between the rat strains. Nonetheless, the expression of four genes went up only in Wistar rats (*Nedd4l*, *Mapk8*, *Runx1*, and *Rdx*) and the expression of six genes increased only in OXYS rats (*Cldn19*, *Ppp2ca*, *Slc9a3r1*, *Was*, *Prkacb*, and *Tuba1b*). At the same time, the expression of six genes declined only in Wistar rats (*Cldn6*, *Pard6a*, *Pard6b*, *Llgl2*, *Erbb2*, and *Arhgef18*), and the expression of four genes diminished only in OXYS rats (*Jam3*, *Bves*, *Pard3*, and *Rap1a*). As for interstrain differences (Appendix A; Figure 3), in the hippocampus of OXYS rats, there were eight DEGs of the TJ signaling pathway at PND3 (three genes with upregulated expression: *Scrib*, *Arpc3*, and *Myl6*; and five genes with downregulated expression: *Actr2*, *Prkacb*, *Myl9*, *Synpo*, and *Tuba1b*), and 21 DEGs at PND10 (eight genes with upregulated expression: *Amotl2*, *Llgl1*, *Scrib*, *Sympk*, *Actg1*, *Src*, *Myh14*, and *Actn1*; and 13 genes with downregulated expression: *Cldn10*, *Cldn11*, *Mpp5*, *Mpdz*, *Amot*, *Hspa4*, *Ezr*, *Rdx*, *Actr2*, *Arpc5*, *Rock1*, *Rab8b*, and *Tuba1b*).

In the frontal cortex of OXYS and Wistar rats, 77 genes belonging to the TJ signaling pathway changed their expression in the same way from PND3 to PND10 (Appendix A): the mRNA levels of 47 genes increased and of 30 genes decreased. Meanwhile, the expression of four genes rose only in Wistar rats (*Cldn3*, *Ppp2ca*, *Dlg2*, and *Cftr*), and the expression of 12 genes rose only in OXYS rats (*Cldn11*, *F11r*, *Pard6a*, *Pard6b*, *Tjp3*, *Tjp1*, *Cgnl1*, *Arpc1b*, *Whamm*, *Rab13*, *Stk11*, and *Marveld2*). Furthermore, the expression of 12 genes diminished only in Wistar rats (*Cldn23*, *Cldn7*, *Bves*, *Pard6g*, *Mpp4*, *Amotl1*, *Llgl1*, *Tjap1*, *Nedd4*, *Ccnd1*, *Map2k7*, and *Prkaa2*), and the expression of 11 genes diminished only in OXYS rats (*Rac1*, *Ppp2r1b*, *Ppp2r2a*, *Cgn*, *Hspa4*, *Rdx*, *Arpc1a*, *Arpc5*, *Rock1*, *Myl12b*, and *Rap1a*). The *Arpc3* mRNA level changed with age in different directions when Wistar and OXYS rats were compared: in Wistar rats, its expression increased from PND3 to PND10, whereas in OXYS rats, this parameter decreased. As for interstrain differences (Appendix A; Figure 3), at PND3, there were seven DEGs with upregulated expression (*Hspa4*, *Arpc3*, *Rock1*, *Myl6*, *Myl12b*, *Marveld2*, and *Tuba1a*) and 13 DEGs with downregulated expression (*F11r*, *Pard6a*, *Ppp2r2c*, *Tjap1*, *Cgnl1*, *Slc9a3r1*, *Ezr*, *Arpc1b*, *Stk11*, *Myl9*, *Synpo*, *Actn1*, and *Tjp2*) from the TJ signaling pathway in the frontal cortex of OXYS rats compared to Wistar rats. The number of these DEGs decreased by PND10: there were only four genes with overexpression (*Nedd4*, *Ybx3*, *Prkaa2*, and *Marveld2*) and seven genes with underexpression (*Cldn10*, *Amot*, *Dlg1*, *Arhgef2*, *Jun*, *Ezr*, and *Actr2*); in the frontal cortex, the mRNA level of *Marveld2* was elevated and that of *Ezr* was reduced in OXYS rats compared to Wistar rats both at PND3 and PND10. At PND3, two genes—*Arpc3* and *Myl6*—had higher expression, and two other genes—*Myl9* and *Synpo*—had lower expression in OXYS rats than in Wistar rats both in the hippocampus and in the frontal cortex. At PND10, four genes (*Cldn10*, *Amot*, *Ezr*, and *Actr2*) had lower expression both in the hippocampus and frontal cortex of OXYS rats compared to Wistar rats.

#### 2.2.2. Adherens Junctions

Closest to the basolateral membrane, adherens junction proteins form homophilic endothelial-to-endothelial contacts [23]. Adherens junctions modulate receptor signaling and regulate the transendothelial migration of lymphocytes, monocytes, and neutrophils [17].

We demonstrated that from PND3 to PND10, 48 of 72 genes belonging to the “adherens junction” pathway changed their expression in the hippocampus of OXYS rats (28 DEGs were upregulated, and 20 DEGs were downregulated), and 54 genes changed their expression in the hippocampus of Wistar rats (30 DEGs were upregulated, and 24 DEGs were downregulated) (Appendix A). Among them, genes *Was* and *Nlk* increased whereas *Pard3*, *Ctnnb1*, and *Fgfr1* decreased their expression only in OXYS rats; by contrast, four genes (*Nectin4*, *Rac2*, *Tgfbr1*, and *Smad4*) increased and seven genes (*Ctnna1*, *Acp1*, *Ptpn1*, *Tcf7l1*, *Erbb2*, *Mapk3*, and *Map3k7*) decreased their expression only in Wistar rats. Regarding interstrain differences (Appendix A; Figure 4), in the hippocampus of OXYS rats at PND3, the *Crebbp* gene had higher expression, and four genes (*Ssx2ip*, *Cdh1*, *Ptprm*, and *Map3k7*) had lower expression relative to Wistar rats. At PND10, seven genes (*Nectin2*, *Src*, *Actn1*, *Actg1*, *Ptprf*, *Mapk3*, and *Crebbp*) had higher expression, and three genes (*Ssx2ip*, *Ptpn6*, and *Tgfbr1*) had lower expression as compared to Wistar rats.

In the frontal cortex (Appendix A), from PND3 to PND10, there were 45 DEGs in Wistar rats (19 DEGs were upregulated, and 26 DEGs were downregulated) and 51 DEGs in OXYS rats (26 DEGs were upregulated, and 25 DEGs were downregulated). Among them, two genes increased their expression (*Wasl* and *Wasf1*) and five genes decreased their expression (*Wasf3*, *Sorbs1*, *Ptprf*, *Snai1*, and *Map3k7*) only in Wistar rats; nine genes increased their expression (*Tjp1*, *Cdh1*, *Tcf7l1*, *Lef1*, *Egfr*, *Fgfr1*, *Snai2*, *Tgfbr2*, and *Smad3*) and four genes decreased their expression (*Rac1*, *Ctnnb1*, *Acp1*, and *Ep300*) only in OXYS rats. Meanwhile, at PND3, the expression of three genes (*Lmo7*, *Tgfbr1*, and *Crebbp*) was excessive and the expression of 12 genes (*Nectin1*, *Nectin2*, *Rac2*, *Actn1*, *Cdh1*, *Ctnna1*, *Ptprm*, *Ptpn6*, *Tcf7l1*, *Lef1*, *Smad3*, and *Map3k7*) was low in OXYS rats compared to Wistar rats; at PND10, the expression of five genes (*Ptprf*, *Insr*, *Yes1*, *Smad4*, and *Crebbp*) was upregulated and that of two genes (*Rac2* and *Ptprm*) was downregulated in OXYS rats compared to Wistar rats (Appendix A; Figure 4).

#### 2.2.3. Gap Junctions

Gap junctions form hemichannels between endothelial cells, thus enabling endothelial intercellular communications [17]. Furthermore, brain endothelial gap junctions also help to maintain TJ integrity [24].

As for the gap junction signaling pathway, 61 of 87 genes changed their expression from PND3 to PND10 in the hippocampus of OXYS and Wistar rats (Appendix A). Among them, 37 DEGs with upregulated expression and 17 DEGs with downregulated expression were common between the two strains. Meanwhile, the expression of four genes (*Sos1*, *Nras*, *Map3k2*, and *Tubb2a*) increased only in Wistar rats, and expression of six genes (*Pdgfc*, *Tuba1b*, *Adrb1*, *Drd1*, *Prkacb*, and *Gucy1a2*) increased only in OXYS rats; there were three DEGs (*Mapk3*, *Adcy7*, and *Gucy1a1*) with downregulated expression only in Wistar rats, and only *Itpr3* diminished its expression in OXYS rats. As to interstrain differences (Appendix A; Figure 5), we observed nine DEGs in OXYS rats at PND3 (upregulated: *Htr2c*; downregulated: *Gja1*, *Pdgfb*, *Pdgfra*, *Pdgfrb*, *Tuba1b*, *Tubb4a*, *Prkacb*, and *Gucy1b1*) and 13 DEGs at PND10 (upregulated: *Gja1*, *Map2k2*, *Mapk3*, *Src*, *Tubb3*, *Gnas*, *Adcy5*, and *Adcy8*; downregulated: *Pdgfra*, *Tuba1b*, *Tubb4a*, *Itpr1*, and *Gucy1b1*). Meanwhile, in OXYS rats, the level of *Gja1* mRNA at PND3 was lower and at PND10 was higher in comparison with Wistar rats.

Regarding the gap junction signaling pathway in the frontal cortex (Appendix A), from PND3 to PND10, 54 genes altered their expression in Wistar rats (32 DEGs with an increase in expression and 22 DEGs with a decrease in expression), and there were 61 DEGs in OXYS rats (39 DEGs with elevated expression and 22 DEGs with underexpression). Among them, the *Tubb4a* gene elevated its expression only in Wistar rats, and eight genes increased their expression only in OXYS rats (*Lpar1*, *Pdgfa*, *Egfr*, *Tjp1*, *Adcy1*, *Adcy4*, *Plcb3*, and *Gucy1b2*); the mRNA levels of three genes declined only in Wistar rats (*Gnai2*, *Map3k2*, and *Tubb6*), and the mRNA levels of three other genes decreased only in OXYS rats (*Map2k5*, *Drd1*, and *Itpr1*). As for interstrain differences (Appendix A; Figure 5), there were 12 DEGs at PND3 (upregulated: *Sos1*, *Tuba1a*, *Drd1*, and *Gnas*; downregulated: *Gja1*, *Lpar1*, *Pdgfb*, *Pdgfrb*, *Tubb4a*, *Adcy4*, *Gna11*, and *Plcb3*) and seven DEGs at PND10 (upregulated: *Tubb2b*, *Grm5*, *Gnaq*, and *Prkcb*; downregulated: *Tubb4a*, *Adcy2*, and *Gucy1b1*) in OXYS rats compared to Wistar rats; the expression of the *Tubb4a* gene was lower in OXYS rats at both ages. At PND3, four genes (*Gja1*, *Pdgfb*, *Pdgfrb*, and *Tubb4a*) of the gap junction signaling pathway were underexpressed both in the hippocampus and frontal cortex of OXYS rats compared to Wistar rats. It is worth pointing out that at PND10, the mRNA levels of genes *Tubb4a* and *Gucy1b1* were lower in the hippocampus and higher in the frontal cortex of OXYS rats compared to Wistar rats.

### 2.3. Permeability of the BBB in the Hippocampus and Frontal Cortex of OXYS and Wistar Rats

We found that in the hippocampus, BBB permeability decreased during its formation and maturation of barrier properties (factorial ANOVA, factor “age”: F_2,23_ = 25.8, *p* < 0.001; Figure 6a). At the same time, OXYS and Wistar rats manifested different dynamics of the decrease in BBB permeability (factorial ANOVA, interaction between factors “age” and “strain” [i.e., genotype]: F_2,23_ = 4.4, *p* < 0.03). Indeed, BBB permeability in the hippocampus of Wistar rats did not change during the second week of life (from PND7 to PND14) and diminished from PND14 to PND20 (Tukey’s test: *p* < 0.03); in the meantime, in OXYS rats, this parameter significantly decreased from PND7 to PND14 (Tukey’s test: *p* < 0.0001) and did not change subsequently. As a consequence, hippocampal BBB permeability was lower in OXYS rats than in Wistar rats at PND14 (Tukey’s test: *p* < 0.05).

In the frontal cortex, BBB permeability also declined from PND7 to PND20 (factorial ANOVA, factor “age”: F_2,48_ = 14.0, *p* < 0.0001; Figure 6b); however, the age-related dynamics of these changes differed between OXYS and Wistar rats (factorial ANOVA, interaction between factors “age” and “genotype”: F_2,48_ = 3.8, *p* < 0.03). Indeed, Wistar rats showed only a marginally significant decrease in BBB permeability from PND7 to PND20 (Tukey’s test: *p* = 0.06), whereas in OXYS rats, this parameter decreased from PND7 to PND14 significantly (Tukey’s test: *p* < 0.0002), and this decline continued during the third week of life (from PND14 to PND20; Tukey’s test: *p* < 0.04). Finally, BBB permeability in the frontal cortex of OXYS rats was significantly lower relative to Wistar rats at PND20 (Tukey’s test: *p* < 0.01).

Because we found more interesting differences in the dynamics of age-related changes of BBB permeability in the hippocampus, afterwards, we studied TJs and the ultrastructure of hippocampal capillaries.

### 2.4. Occludin and Claudin-5 Levels in the Hippocampus of OXYS and Wistar Rats

The occludin level in the hippocampus was relatively stable from birth to PND20 and did not differ between OXYS and Wistar rats (Figure 7). By contrast, this was not the case for the hippocampal claudin-5 level (Figure 7), which changed significantly with age (factorial ANOVA, factor “age”: F_3,32_ = 5.7, *p* < 0.003). In the hippocampus of Wistar rats, the claudin-5 level rose significantly during the first 2 weeks of life (from PND0 to PND14; Tukey’s test: *p* < 0.02) and then tended to decrease until PND20 (Tukey’s test: *p* = 0.06). At the same time, OXYS rats did not demonstrate any significant age-related changes in claudin-5 levels.

### 2.5. Ultrastructural Organization of Capillaries in the CA1 Hippocampal Area of OXYS and Wistar Rats

Ultrastructural analysis of these capillaries indicated that the average length of endothelial TJs did not differ between OXYS and Wistar rats at PND14 (Figure 8a); however, this parameter was significantly greater in OXYS rats than in Wistar rats at PND20 (Mann–Whitney *U* test: *p* < 0.001). Nonetheless, it should be pointed out that the proportion of TJs among all intercellular contacts did not differ between OXYS and Wistar rats at PND14 and PND20 (Figure 8b). In addition, at PND14, hippocampal endotheliocytes of OXYS rats had a lower content of mitochondria relative to Wistar rats (ANOVA analysis: *p* < 0.03; Figure 8c).

In terms of pericytes (Figure 8d,e), at PND7, we detected them in almost 100% of analyzed capillaries in both Wistar and OXYS rats. At PND14, 96% of examined capillaries in Wistar rats and only 29% of examined capillaries in OXYS rats were surrounded by pericytes. At PND20, 95% of capillaries in Wistar rats and 100% of capillaries in OXYS rats contained pericytes. Indeed, we noted that at PND14, the pericyte perivascular area was significantly smaller in the *cornu ammonis* 1 (CA1) hippocampal area of OXYS rats compared to Wistar rats (Mann–Whitney *U* test: *p* < 0.0004); however, this difference disappeared by PND20.

### 2.6. Astrocyte Density in the CA1 Hippocampal Area of OXYS and Wistar Rats

Astrocyte end-foot processes are an important component of the BBB, and reduced astrocyte density may lead to a disturbance of BBB integrity. Previously, we have reported that astrocyte density is lower in the CA1 hippocampal area of OXYS rats compared to Wistar rats at PND7 [13].

Here we analyzed astrocyte density in the CA1 area of the hippocampus in OXYS and Wistar rats at PND14 and PND20 (Figure 9). We showed that this parameter decreased with age (Spearman’s rank correlation: R = −0.60, *p* < 0.02). This decline was significant in Wistar rats: indeed, astrocyte density in the CA1 hippocampal area diminished from PND14 to PND20 (Mann–Whitney *U* test: *p* < 0.03). The decrease, however, was absent in OXYS rats; namely, at PND20, there was only marginally higher astrocyte density in OXYS rats than in Wistar rats (Mann–Whitney *U* test: *p* = 0.06).

### 2.7. Illustration of Vessel Density in Different Brain Areas of OXYS and Wistar Rats

As the last step of our work, we obtained illustrative images of brain vessels in different brain areas of OXYS and Wistar rats at PND7, PND14, and PND20. CD31 protein was used as a blood vessel marker.

We found that vessel density was lower in the hippocampus, frontal cortex, and cerebellum of OXYS rats compared to Wistar rats at PND7 (Figure 10a–c); by PND14, these differences disappeared (Figure 10a,b).

## 3. Discussion

In the present study, we investigated whether the efficiency of BBB formation in the hippocampus and prefrontal cortex of OXYS rats would influence the delays in the completion of postnatal brain maturation and thus contribute to the development of AD-like pathology in these animals.

It is widely accepted that a preterm birth or a low weight at birth lead to altered vascular function later in life: this means hypertension caused by impairments in vessel formation, altered expression of extracellular-matrix molecules, oxidative stress, and inflammation in infants [25,26,27]. Vascular dysfunction is one of the major factors contributing to neurodegenerative diseases including AD [28]. The importance of the vascular component in brain aging is emphasized by Schaffenrath and colleagues [29], who identified reduced vascular density in the hippocampus of aged short-lived mice compared to normal-lived animals. A predisposition to neurodegenerative diseases with vascular involvement may arise at an early age, especially in preterm infants. Investigation of blood vasculature formation in this context is of particular interest. For this purpose, in the present study, we used OXYS rats, for which retardation of pregnancy and low weight at birth have been documented [30]. Here we researched the maturation of the BBB in the hippocampus and frontal cortex of OXYS rats and of their parental strain (Wistar rats) during the first 3 weeks of life.

We showed that in Wistar rats, the establishment of TJs (evidenced by an increase in the level of the claudin-5 protein in the first 2 postnatal weeks) was followed by a decrease in BBB permeability in terms of passive diffusion by the end of the third week of life (Figure 6 and Figure 7). This decrease of BBB permeability after the increase in the claudin-5 level seems to be natural because it is well known that claudin-5 constitutes a size-selective barrier to small molecules (<800 Da, [31]), whereas the molecular weight of Evans blue dye is 67 kDa [32]. Meanwhile, the decrease of astrocyte density in the CA1 area of the hippocampus during the third week of life (which we observed in Wistar rats) may point to completion of active gliogenesis and a redistribution of newly generated astrocytes to their adult destination points. These results are in line with our previously obtained data on astrocytic density in the dentate gyrus [30].

As for OXYS rats, a delay in the formation of pial vessels in OXYS rats during the first two weeks of life was demonstrated previously [33]. Indeed, mitotic activity of the endothelium (MAE) of pial arteries, arterioles, and veins was lower in OXYS rats than in Wistar rats at PND4; at PND12, MAE was still lower in the pial arteries and veins of OXYS rats, but this parameter was higher in venules; at PND30, MAE was higher in the pial arterioles and venules of OXYS rats. The findings of the current work support these results: indeed, we demonstrated a decrease in the amount of blood vessels in various brain structures of OXYS rats at PND7. It is important to state that the number of brain blood vessels did not differ between OXYS and Wistar rats at PND14, implying extensive growth of vessels during the second week of life in OXYS rats, possibly resulting in the dramatic decrease in the percentage of capillaries covered with pericytes in the hippocampus of OXYS rats at PND14 because pericytes take their position late in vascular formation [34]. Finally, given that extracerebral arteries at the surface of the brain, such as pial arteries, are the main site of blood flow control in the brain [35], we can hypothesize that the lower amounts of Evans blue dye in the hippocampus of OXYS rats at PND14 reflect not only a decrease in BBB permeability but also a decline in cerebral blood flow. Given that the high-flow condition leads to increased levels of adherens junction and gap junction proteins in endotheliocytes [36], the decreased expression of their genes at PND3 in OXYS rats may indirectly point to lower cerebral blood flow already at an early age.

As for the mitochondrial network and total mitochondrial mass, they are larger in the brain endothelium than in endothelial cells from other organs. This excess is a result of high metabolic functions of the BBB and of the numerous transport systems it supports [37]. In the present paper, we found a decreased mitochondrial content in the hippocampal endotheliocytes of OXYS rats at PND14 (Figure 8c). Thus, the energy demands of endotheliocytes may remain unsatisfied.

Analysis of gene expression revealed that in OXYS rats compared to Wistar rats, the number of DEGs associated with the BBB and junction formation was higher in the frontal cortex at PND3 and in the hippocampus at PND10. Therefore, in the discussion section, we focus on the frontal cortex at PND3 and on the hippocampus at PND10.

From the data of the RNA-Seq analysis, we could conclude that in the prefrontal cortex of OXYS rats at PND3, a pericyte deficit could be present. Indeed, it is known that pericyte proliferation during angiogenesis in the CNS depends on PDGF-B/PDGFRb signaling [38,39], and we revealed that the expression of both *Pdgfb* and its receptor *Pdgfrb* is low in the prefrontal cortex of OXYS rats compared to Wistar rats at PND3. Moreover, loss-of-function mutations in the *PDGFB* gene within the endothelium or in the *PDGFRB* gene within pericytes can cause primary familial brain calcification, which is characterized by pericyte loss and neurological symptoms including motor and cognitive impairments [40,41]. Additionally, astrocyte laminins—laminin-111 and laminin-211—regulate pericyte differentiation and maintain the expression of TJ proteins [42]. We noticed underexpression of two (*Lama1* and *Lamc1*) of three subunits of laminin-111 in the frontal cortex of OXYS rats compared to Wistar rats. As for basement membrane formation, in the frontal cortex of OXYS rats at PND3, the levels of *Col4a1* and *Col4a2* were lower relative to Wistar rats. These genes encode major basement membrane collagen proteins COL4A1 and COL4A2, and their mutations cause cerebrovascular diseases [43]. Another feature of BBB development in the frontal cortex of OXYS rats concerns transporters. It is known that brain endothelium-specific or brain endothelium-enriched transporters are often indispensable for BBB barrier genesis [17]. We report here that at PND3, the mRNA levels of *Slc2a1* and *Mfsd2a* are decreased, and the mRNA level of *Slc16a2* is increased in the frontal cortex of OXYS rats compared to Wistar rats. *Slc2a1* encodes the GLUT1 protein, which is the major glucose transporter through the BBB [44]; *Mfsd2a* codes for a sodium-dependent lysophosphatidylcholine symporter supplying the brain with an essential circulating omega-3 fatty acid [45] and is required for proper BBB development and functional integrity [46]; *Slc16a2* encodes a transporter (MCT8) of the T3 thyroid hormone [47]. Thus, the expression of transporters of essential supplies was low, and the expression of the hormone transporter was high in the prefrontal cortex of OXYS rats at PND3. It is important to point out that all the observed changes in gene expression within the frontal cortex of OXYS rats disappeared at PND10. This observation may reflect the involvement and possible exhaustion of compensatory mechanisms, which took place in OYXS rats during early development. This early exhaustion of compensation mechanisms may result in a progression of neurodegeneration observed in OXYS rats later in life.

As for gene expression in the hippocampus at PND10, we documented overexpression of gene coding for proteins that are the components of adherens junctions (genes *Ptprf* and *Marcksl1*) and gap junctions (*Gja1*) in OXYS rats; these alterations may lead to the aforementioned decrease of BBB permeability at PND14. Meanwhile, we revealed not only an overexpression of a matrix metalloproteinase protein family member (*Mmp15* gene) but also an underexpression of a negative regulator for matrix metalloproteinase 9 (*Reck* gene). Taken together, these findings point to lower stability of the basement membrane in the hippocampus of OXYS rats: matrix metalloproteinases weaken the barrier properties of the extracellular matrix and reduce cell–cell adhesion by destroying intercellular junctions [48], whereas RECK limits basement membrane degradation [49]. Moreover, we found an imbalance in the expression of mitochondrial proteins: elevated levels of mitochondrial transporters (genes *Slc25a10* and *Sfxn1*) and reduced levels of enzymes (genes *Maob* and *Gpd2*) and of apoptotic factors (genes *Bnip3* and *Higd1a*). The observed alterations of expression of mitochondrial genes in the hippocampus of OXYS rats at PND10 may eventually result in decreased mitochondrial content in the endothelial cells at PND14.

Altogether, our results revealed specific features of vascular development in the hippocampus and frontal cortex of OXYS rats in the first 3 weeks of life. These features include a delay of vascularization and of subsequent pericyte coating of blood vessels, accompanied by an accelerated decline of BBB permeability. It must be emphasized that almost all the observed differences from Wistar rats disappeared by PND20. Nevertheless, these features (characteristic of the early postnatal period) may have long-term consequences and contribute to the neurovascular dysfunction observed in OXYS rats late in life [50], thereby promoting early development of AD signs.

## 4. Materials and Methods

### 4.1. Animals

The senescence-accelerated OXYS rat strain was developed at the Institute of Cytology and Genetics (ICG), SB RAS (Novosibirsk, Russia), from a Wistar stock [10]. OXYS rats and age-matched Wistar rats were obtained from the Breeding Experimental Animal Laboratory of the ICG SB RAS (Novosibirsk, Russia). The animals were kept under standard laboratory conditions (22 ± 2 °C; 60% relative humidity; a 12 h light/12 h dark cycle), had *ad libitum* access to standard rodent feed (PK-120-1, Laboratorsnab, Ltd., Moscow, Russia) and water. The study was conducted according to Directive 2010/63/EU of the European Parliament and of the European Council of 22 September 2010 and was approved by the Commission on Bioethics at the ICG SB RAS (decision # 34 of 15 June 2016), Novosibirsk, Russia.

### 4.2. RNA-Seq Analysis

The RNA-Seq data from the hippocampus and prefrontal cortex of OXYS and Wistar rats at PND3 and PND10 (*n* = 3 per strain and age) had been obtained earlier, as described elsewhere [51]. In brief, the hippocampus and prefrontal cortex were fixed into RNALater (cat. # AM7020, Thermo Fisher Scientific, Waltham, MA, USA) and frozen. Frozen rat tissues were lysed with TRIzol Reagent (cat. # 15596–018, Thermo Fisher Scientific), and total RNA was isolated. RNA quality and quantity were evaluated on an Agilent Bioanalyzer (Agilent, Santa Clara, CA, USA). More than 40 million single-end reads 50 bp long were obtained for each sample of RNA, by Illumina nonstranded sequencing on an Illumina GAIIx instrument at the Genoanalitika Lab, Moscow, Russia in accordance with standard Illumina protocols using mRNA-Seq Sample Preparation Kit (Illumina Inc., San-Diego, CA, USA).

The sequencing data were preprocessed using the Cutadapt tool (https://cutadapt.readthedocs.io; version 2.2; accessed on 22 October 2023) to remove adapters and low-quality sequences. The resulting reads were mapped onto the Rnor_5.0 reference genome assembly in the TopHat2.1.1 software (https://ccb.jhu.edu/software/tophat/; accessed on 22 October 2023). The data were then converted into gene count tables by means of ENSEMBL and RefSeq gene annotation data. The resulting tables were subjected to analysis of differential gene expression in the DESeq2 software (https://bioconductor.org/packages/release/bioc/html/DESeq2.html; Bioconductor version: Release (3.17); accessed on 22 October 2023). Genes with Padj < 0.05 were designated as differentially expressed.

Pathway analysis of the DEGs was conducted by means of the WEB-based Gene Set Analysis Toolkit (WebGestalt 2019: gene set analysis toolkit with revamped UIs and APIs) using KEGG Pathways (https://www.genome.jp/kegg/ accessed on 22 October 2023).

### 4.3. Staining of Brain Structures Using Evans Blue Dye

BBB permeability of the hippocampus and prefrontal cortex was studied in OXYS and Wistar rats at PND7, PND14, and PND20 (*n* = 6 per strain and age). Evans blue is a synthetic dye with a molecular weight of 67,000 Da [32]. Evans blue was injected intraperitoneally (a 2% solution in saline [0.9% NaCl] at 3 mL per kg of body weight); after 24 h, the animals were subjected to avertin anesthesia, and intracardiac perfusion with saline (0.9% NaCl) was performed to wash the dye out of the body. Then, the animals were decapitated, and brain structures were carefully excised, placed in tubes, and frozen in liquid nitrogen. To determine the concentration of Evans blue dye in a tissue, hippocampal and frontal-cortex samples were weighed, homogenized in trichloroacetic acid in saline (sample:trichloroacetic acid:saline at 1:2:2), and then centrifuged at 10,000× *g* at 4 °C for 20 min. Supernatants were separated and kept at 4 °C for 24 h for better precipitation. Then, the samples were centrifuged at 10,000× *g* at 4 °C for 20 min again, and the supernatants were collected. Next, 30 μL of the supernatant of each sample was placed into an individual well of a 96-well plate and diluted with 90 μL of 96% ethanol. Fluorescence was measured on a CLARIOstar Plus microplate reader (BMG Labtech, Ortenberg, Germany) with an absorption/emission range of 620/680 nm. The concentration of Evans blue was calculated from a calibration curve constructed with various dye concentrations (0; 0.75; 1.5; 3; 6; 12; 25; and 50 μg/mL).

### 4.4. Western Blot Analysis

Male pups of the OXYS and Wistar strains were decapitated at PND0, PND7, PND14, and PND20 (*n* = 6 per strain and age); brains were carefully excised, and hippocampi were placed in tubes and frozen in liquid nitrogen. The hippocampi were stored at −70 °C until further processing.

The hippocampal samples were weighed, and then lysis buffer (RIPA pH 7.6: 50 mM Tris-HCl pH 8.0, 150 mM NaCl, 1% of Triton X-100, 0.1% of sodium dodecyl sulfate [SDS], and 0.5% of sodium deoxycholate) and protease and phosphatase inhibitors (cat. ## P8340, P0044, and P5726, Sigma–Aldrich, St. Louis, MO, USA) were added at the following ratio: 1 mg of a sample/10 μL of lysis buffer/0.1 μL of protease inhibitors/0.05 μL of phosphatase inhibitors. Then, the hippocampal samples were homogenized and centrifuged at 12,000× *g* at 4 °C for 30 min; the supernatant, which contained soluble proteins, was separated. The total concentration of proteins was determined by spectrophotometry using a specialized kit (Pierce BCA Protein Assay Kit, Thermo Fisher Scientific). After that, the samples were diluted by loading Laemmli sample buffer (5× Laemmli sample buffer: 2.5% of SDS, 12.5% of β-mercaptoethanol, 45% of glycerol, 17.5% of 0.5 M Tris-HCl pH 6.8, 0.25% of bromophenol blue, and 22.25% of bidistilled H_2_O) at a 4:1 ratio, and were incubated at 95 °C for 10 min. Approximately 50 μg of the total protein from the hippocampal samples was resolved by SDS-PAGE (a 10% gel for occludin, MW = 59 kDa; a 15% gel for claudin-5, MW = 18 kDa) in Tris-glycine running buffer (25 mM Tris base, 190 mM glycine, and 0.1% of SDS) and transferred to a nitrocellulose membrane (Hybond–CExtra, cat. # RPN303D, GE Healthcare, Chicago, IL, USA). The membrane was blocked with 3% bovine serum albumin (BSA; Sigma-Aldrich, St. Louis, MO, USA) in Tris-buffered saline supplemented with 0.1% Tween 20 (TBS-T) for 1 h at room temperature (RT), and incubated overnight at 4 °C with an antibody to occludin (1:1000 dilution, cat. # ab167161, Abcam, Cambridge, MA, USA) or to claudin-5 (1:1000, cat. # AF5216, Affinity Biosciences, Cincinnati, OH, USA) as well as to β-actin (1:5000 cat. # ab6276, Abcam) as an internal loading control. Secondary antibodies were HRP-conjugated anti-IgG antibodies (1:5000 cat. ## ab97051 and ab97046, Abcam; 1:10,000 cat. # ab6276, Abcam). After incubation with the respective secondary antibodies, chemiluminescence signals were measured and scanned, and the intensity of the emission bands was quantified using the ImageJ software (NIH, Bethesda, MD, USA; Release 1.53v 21).

### 4.5. Electron-Microscopic Examination

For this purpose, the hippocampal samples of OXYS and Wistar rats at PND7, PND14, and PND20 (*n*  =  3 per strain and age) were fixed with 2.5% glutaraldehyde in sodium cacodylate buffer (pH 7.2) for 1 h, rinsed with 0.1 M sodium cacodylate buffer, and postfixed in 1% osmium tetroxide in the same buffer for 1 h. After that, the samples were washed with water and incubated in a 1% aqueous solution of uranyl acetate in the dark at RT for 1 h. The samples were then dehydrated using a graded series of ethanol–acetone mixtures and were embedded in Agar 100 resin.

Ultrathin slices were prepared and double-stained with uranyl acetate and lead citrate and then examined under a transmission electron microscope (JEM 100 SX; Jeol, Tokyo, Japan) at the Interinstitutional Centre for Microscopic Analysis of Biological Objects, the Institute of Cytology and Genetics, SB RAS.

For the evaluation of morphological features, capillaries in the pyramidal layer of the CA1 area were identified in the electron micrographs (at least seven cross-sections of capillaries per group for quantitative analysis at PND14 and PND20; 3–5 cross sections of capillaries for demonstration at PND7). Ultrathin sections were studied and photographed by means of an electron microscope JEM-7A.

### 4.6. Immunohistochemistry

Male pups of the OXYS and Wistar strains were decapitated at PND7, PND14, and PND20; their brains were carefully excised, and the hemispheres were separated and immediately fixed in 4% paraformaldehyde in phosphate-buffered saline (PBS) at RT for 48 h, followed by cryoprotection in 30% sucrose in PBS at 4 °C for 48 h. Then, the brains were frozen and stored at −70 °C until further processing. Brain sagittal sections (20 μm thick) from OXYS and Wistar rats (*n* = 4 per strain and age; three technical replicates for each animal) were prepared on a Microm HM-505 N cryostat (Microm, Walldorf, Germany) at −20 °C and transferred onto polysine-glass slides (Menzel-Glaser, Braunschweig, Germany). After serial washes with PBS, the slices were incubated at RT for 15 min in PBS-plus (PBS with 0.1% of Triton X-100) and for 1 h in 3% BSA (cat. # A3294, Sigma-Aldrich) in PBS to permeabilize the tissues and to block nonspecific binding sites, and were then incubated overnight with primary antibodies at 4 °C. The primary antibody to glial fibrillary acid protein (GFAP) (cat. # ab7260, Abcam) or to CD31 (cat. # ab119339, Abcam) was diluted 1:250 with 3% BSA in PBS. After several washes with PBS, the slices were probed with secondary antibodies conjugated with Alexa Fluor 488 (cat. # ab175472, Abcam) in PBS (1:250) for 1 h at RT and next were washed in PBS. The slices were coverslipped with the Fluoroshield mounting medium containing DAPI (cat. # ab104139, Abcam). Negative controls were processed in an identical manner except that a primary antibody was not included. The GFAP and CD31 signals were detected under a microscope with a 40× objective lens (Axioskop 2 plus, Zeiss, Oberkochen, Germany). The microscopy was conducted at the Multi-Access Center for Microscopy of Biological Objects (ICG SB RAS, Novosibirsk, Russia). Identification of the different brain areas was performed according to Paxinos and Watson (Lateral 0.40 to Lateral 0.90 mm) [52]. The total number of astrocytes was determined by means of the ZEN software (Zeiss; Release 3.7). To evaluate cell density, the total number of counted cells was divided by the CA1 area, then averaged in each group of three technical replicates per animal, and presented as the number of cells per 10,000 μm^2^.

### 4.7. Statistics

Statistical analysis was performed in STATISTICA 8.0 software (TIBCO Software Inc., Palo Alto, CA, USA). The Kolmogorov–Smirnov test was used to evaluate the normality of the data; a Z-score of ±3 was used to detect outliers. The data were subjected to two-way ANOVA; the genotype and age were chosen as independent factors. Tukey’s test was applied to significant main effects and interactions in order to assess differences between some sets of means. Correlation analysis was employed to estimate associations between parameters. For nonparametric statistics, the Mann–Whitney *U* test was performed. The data are presented as the mean ± SEM. Differences were considered statistically significant at *p* < 0.05.

## Figures and Tables

**Figure 1 ijms-24-15649-f001:**
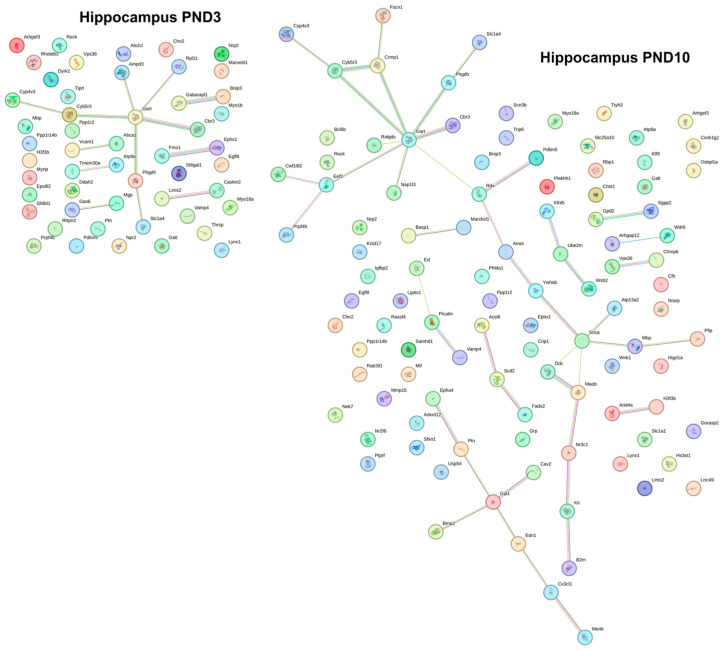
PPI network of proteins encoded by BBB-enriched transcripts with significantly changed expression in the hippocampus of OXYS rats compared to Wistar rats at PND3 and PND10.

**Figure 2 ijms-24-15649-f002:**
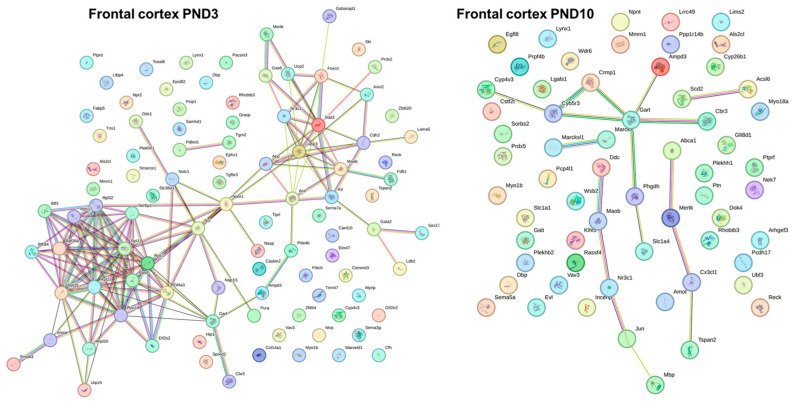
PPI network of proteins encoded by BBB-enriched transcripts with significantly changed expression in the frontal cortex of OXYS rats compared to Wistar rats at PND3 and PND10.

**Figure 3 ijms-24-15649-f003:**
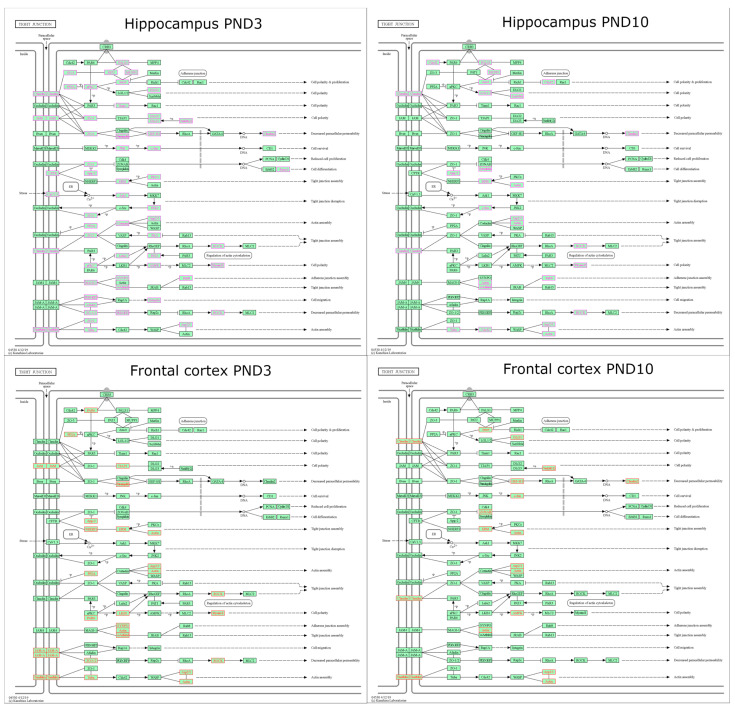
DEGs belonging to the TJ signaling pathway in the hippocampus (magenta) and frontal cortex (red) of OXYS rats at PND3 and PND10.

**Figure 4 ijms-24-15649-f004:**
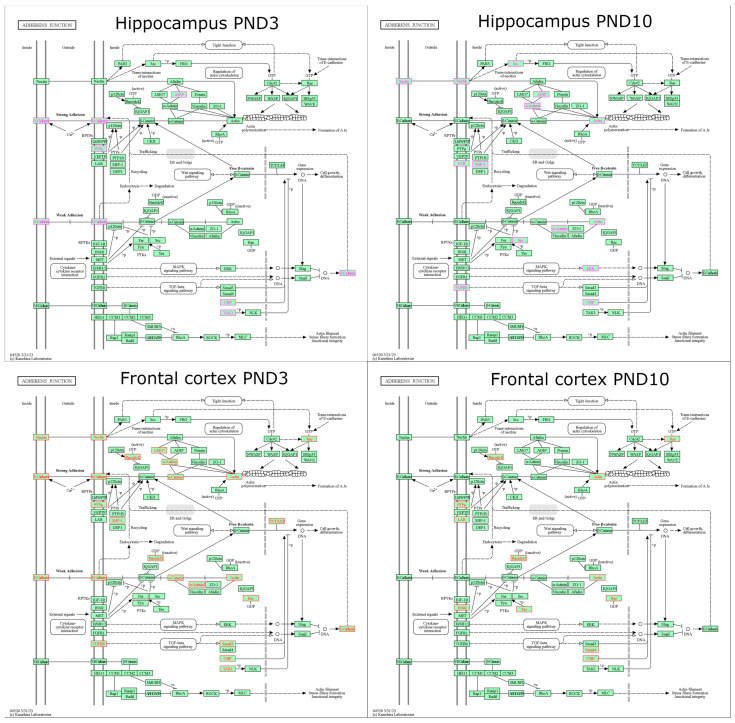
DEGs belonging to the adherens junction signaling pathway in the hippocampus (magenta) and frontal cortex (red) of OXYS rats at PND3 and PND10.

**Figure 5 ijms-24-15649-f005:**
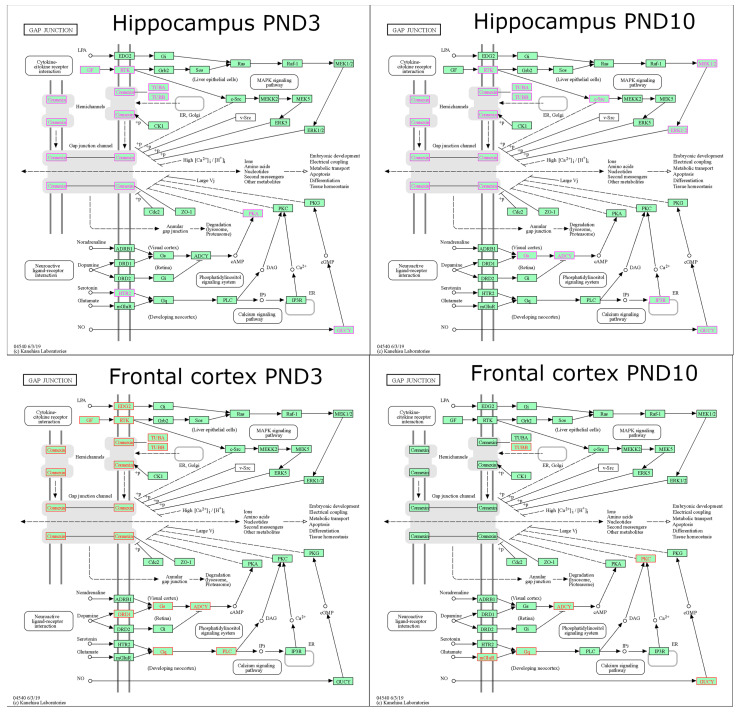
DEGs belonging to the gap junction signaling pathway in the hippocampus (magenta) and frontal cortex (red) of OXYS rats at PND3 and PND10.

**Figure 6 ijms-24-15649-f006:**
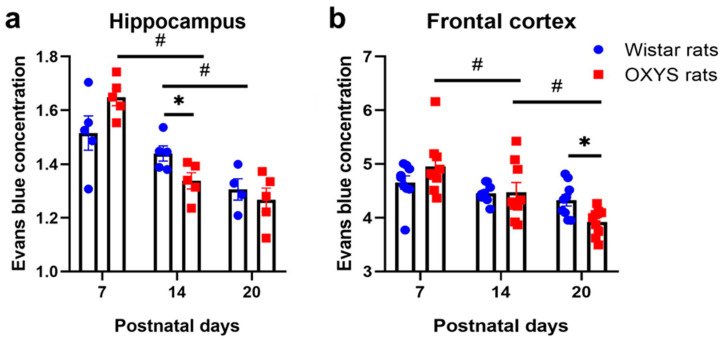
Concentration of Evans blue dye as an indicator of BBB permeability: (**a**) in the hippocampus of OXYS and Wistar rats; (**b**) in the frontal cortex of OXYS and Wistar rats. The data are presented as mean ± standard error of the mean (SEM); * *p* < 0.05 for interstrain differences; ^#^
*p* < 0.05 for age-related differences.

**Figure 7 ijms-24-15649-f007:**
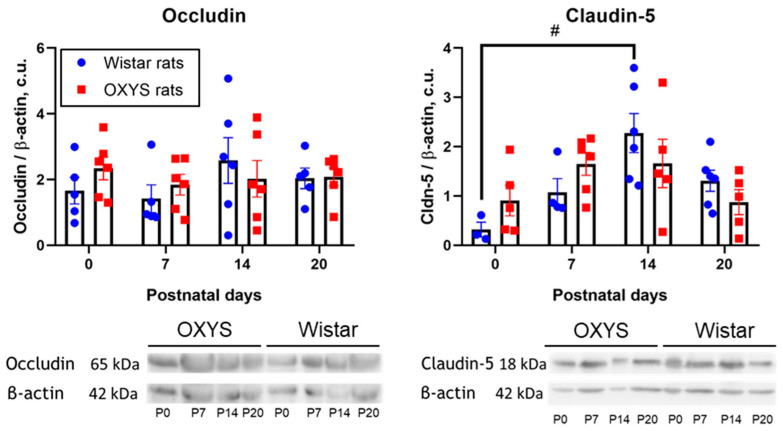
Levels of occludin and claudin-5 in the hippocampus of OXYS and Wistar rats. The data are presented as mean ± SEM; ^#^
*p* < 0.05 for age-related differences; c.u.: conventional units; P: postnatal day.

**Figure 8 ijms-24-15649-f008:**
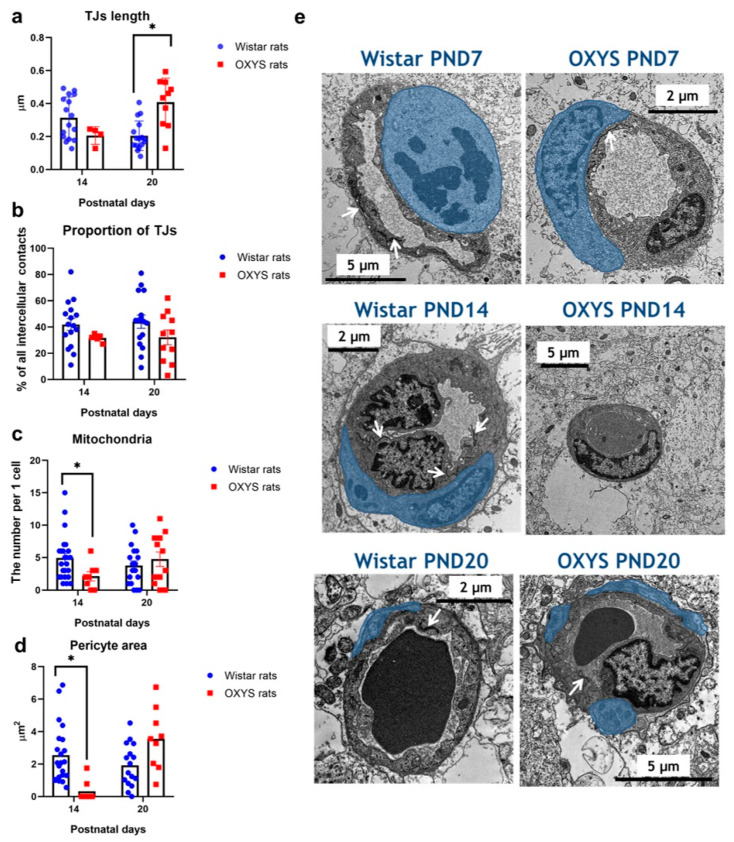
Ultrastructural analysis of capillaries in the CA1 hippocampal area of OXYS and Wistar rats. (**a**) The length of TJs; (**b**) the proportion of TJs among all contacts; (**c**) the number of mitochondria per endothelial cell; (**d**) the area of pericytes. The data are presented as mean ± SEM; * *p* < 0.05 for interstrain differences. (**e**) Representative images of capillaries: arrows indicate TJs; the cytoplasm of pericytes is highlighted in blue. White arrows indicate TJs.

**Figure 9 ijms-24-15649-f009:**
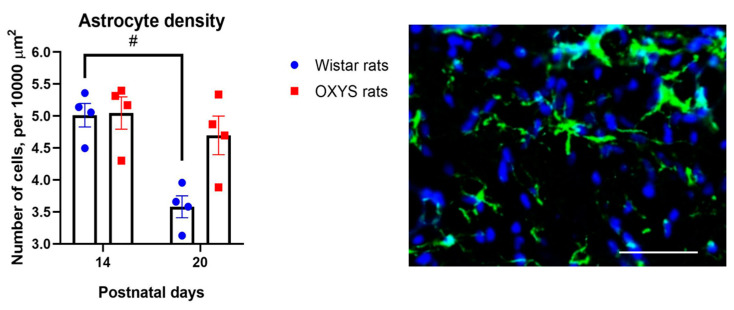
Astrocyte density in the CA1 hippocampal area of OXYS and Wistar rats. The data are presented as mean ± SEM; ^#^
*p* < 0.05 for age-related differences. A representative image of the molecular layer of the CA1 hippocampal area of a Wistar rat (age PND14); glial fibrillary acid protein GFAP (green) indicates astrocytes; 4′,6-diamidino-2-phenylindole (DAPI; blue) denotes cell nuclei; the scale bar is 50 µm.

**Figure 10 ijms-24-15649-f010:**
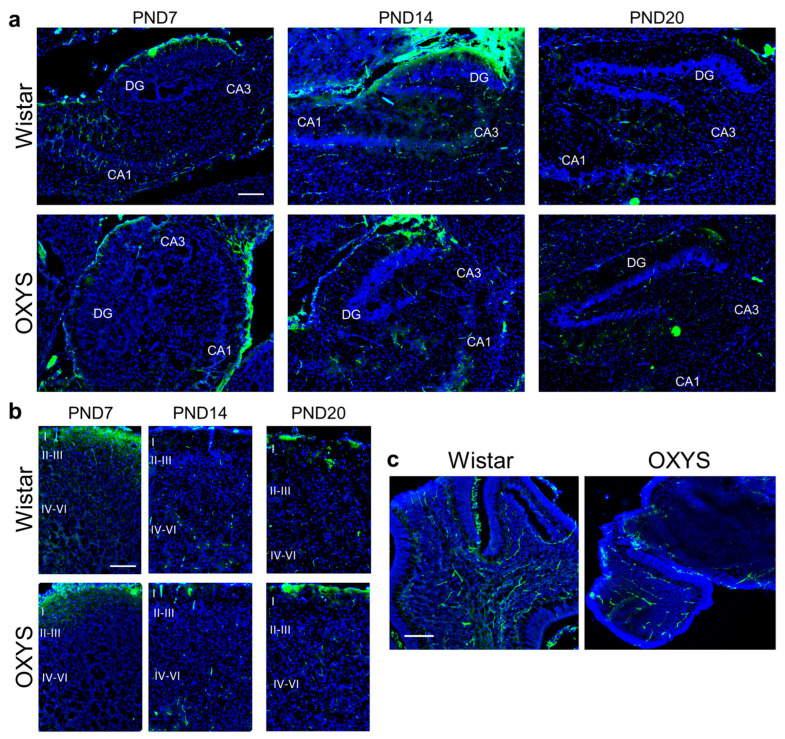
Illustrative images of blood vessels from the brain of OXYS and Wistar rats. (**a**) The hippocampus; (**b**) the frontal cortex; (**c**) the cerebellum at PND7. CA1: *cornu ammonis* 1; CA3, *cornu ammonis* 3; DG: dentate gyrus; I: 1st cortical layer; II–III: 2nd–3rd cortical layers; IV–VI: 4th–6th cortical layers; CD31 (green) denotes blood vessels; DAPI (blue) indicates cell nuclei; the scale bar is 200 µm.

## Data Availability

Raw data are available from the corresponding author upon request.

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
