# Peer review of "Postnatal Maturation of the Blood–Brain Barrier in Senescence-Accelerated OXYS Rats, Which Are Prone to an Alzheimer’s Disease-like Pathology"

_ijms, 2023, doi:10.3390/ijms242115649_

Round 1

Reviewer 1 Report

This study explored early-life vascular dysfunction in Alzheimer's disease (AD) using OXYS rats as a model. The research found altered BBB-associated gene expression and delayed vascularization in the first weeks of life, potentially contributing to neurovascular dysfunction and early AD development in OXYS rats. The research is interesting and relevant to the field. The manuscript contains essential information. However, some explanations are necessary to improve the manuscript. Please find below my suggestions and comments to improve the quality and clarity of the work.

Introduction: 

-       Very well-organized and clear. 

Material and Methods: 

-       On the animal experiments: 

o   Is there any ethical documentation/license used to maintain and work with the animals? The authors should mention it in this section. 

o   Line 438-440- There are too many “and”. The authors should work on rephrasing these sentences better. 

-       On the RNA-Seq analysis: 

o   The authors should describe how the RNA-Seq was performed. Although they reference the paper where the information was described, the authors should at least briefly describe the RNA-Seq method. 

-       Staining of Brain Structures Using the Evans Blue Dye:

o   Line 459: What happens to the sample between the first centrifugation and the second? Is it kept on ice? Room Temperature? The authors say ”Supernatant were separated. After 24h…” it would be important to know where the sample was stored, and why 24h difference between the first and second centrifugation. 

o   Line 464-465: Can the authors describe the calibration curve dilution? It is essential for the study’s reproducibility. 

-       Electron Microscopic Examination: 

o   Line 502: Authors should describe how the samples were prepared, even if other papers are used as reference.

Results: 

-       In section 2.1.: 

o   The authors must describe the analysis performed in the material and methods section. 

o   The authors describe the work done by Daneman and colleagues in 2010 as a reference to the selected BBB genes. I recommend that the authors consult more recent publications discussing BBB-related genes. Some suggestions: PMID: 34321020, PMID: 33563079. 

o   The authors describe the analysis performed to compare the OXYS and Wistar rats and present it in an Excel table in the supplementary file. This information is essential and it should be described/presented in the main article. This information could be better discussed and presented using heat maps or other graphics to make it easier to visualize and enrich the overall results section. 

o   For the BBB-related genes, the authors should show the rationale for the selected genes. The authors focused on DEGs but do not reflect where that family of genes belongs. For example, on line 104 are the tgfb2, Marveld2, and Lrp8 related to which path or structure on the cell? 

o   What is the relationship of all the selected genes on section 2.1. with the BBB? The authors need to be clearer as to why they are mentioning these genes. Supporting figures/tables on the main manuscript would be very beneficial. 

-       In section 2.2.: 

o   Here similar suggestion as in the previous section, the supplementary tables should be added to the main manuscript. To facilitate the reader and the overall understanding of the work. 

o   The authors are describing the genes differentially expressed that are observed in the supplementary tables without reflecting on what that means. Which pathway are they related to?  The authors should consider to improve this section and the previous section to make it more clear and organized for the reader. 

-       In section 2.4.: 

o   The authors have been correctly addressing the genes names using lowercase letters. However, in this section, the authors describe the animal genes using uppercase letters. The authors should uniformize all the manuscript to address the animal genes with lowercase letters. 

-       In section 2.5.: 

o   Line 283: the authors mentioned the mitochondria content from OXYS vs Wistar rats, and the data was not shown. However, in the discussion, the mitochondria content is referred to and discussed. Therefore, since the authors have the data, the manuscript, and the readers would benefit more if that data is shared and explained. 

Discussion: 

-       The authors should significantly improve this section to reflect more on what is observed in the results. An example, there wasn’t any reference to the figures, to make it easier to understand the rationale. All the DEGs observed only a few of them were mentioned/discussed in this section. The authors should consider improving the explanation of the observed results. 

-       Can the authors explain or hypothesize why the gene expression changes on OXYS rats disappeared at PND10? 

Author Response

First of all, we are very grateful to the reviewers for their valuable comments and recommendations. We tried to consider every comment to make the manuscript better.

Material and Methods: 

-       On the animal experiments: 

o   Is there any ethical documentation/license used to maintain and work with the animals? The authors should mention it in this section. 

Thank you for the comment, there is ethical statement at the ‘Institutional Review Board Statement’; however, we added the same statement at the Material and Methods section too.

o   Line 438-440- There are too many “and”. The authors should work on rephrasing these sentences better. 

We reduced the number of “and”.

-       On the RNA-Seq analysis: 

o   The authors should describe how the RNA-Seq was performed. Although they reference the paper where the information was described, the authors should at least briefly describe the RNA-Seq method. 

Thank you for the comment,we added the description to Materials and Methods section.

-       Staining of Brain Structures Using the Evans Blue Dye:

o   Line 459: What happens to the sample between the first centrifugation and the second? Is it kept on ice? Room Temperature? The authors say ”Supernatant were separated. After 24h…” it would be important to know where the sample was stored, and why 24h difference between the first and second centrifugation. 

Thank you for the comment, we clarified the method.

o   Line 464-465: Can the authors describe the calibration curve dilution? It is essential for the study’s reproducibility. 

We added the calibration curve dilution.

-       Electron Microscopic Examination: 

o   Line 502: Authors should describe how the samples were prepared, even if other papers are used as reference.

Thank you for the comment, we added the missing information.

Results: 

-       In section 2.1.: 

o   The authors must describe the analysis performed in the material and methods section. 

We described all the analyses in the Material and Methods section.

o   The authors describe the work done by Daneman and colleagues in 2010 as a reference to the selected BBB genes. I recommend that the authors consult more recent publications discussing BBB-related genes. Some suggestions: PMID: 34321020, PMID: 33563079. 

Thank you for the valuable comment. We chose the work of Daneman and colleagues, because they present the BBB enriched transcripts compared to endothelium of other tissues. Because in our study we used bulk RNA from the brain tissue (hippocampus and frontal cortex), it was of great importance for us to use BBB enriched transcripts rather than endothelial transcriptome. However, we added information from these valuable papers to the Discussion section.

o   The authors describe the analysis performed to compare the OXYS and Wistar rats and present it in an Excel table in the supplementary file. This information is essential and it should be described/presented in the main article. This information could be better discussed and presented using heat maps or other graphics to make it easier to visualize and enrich the overall results section. 

We appreciate your comment, we tried to make the data more well-presented.

o   For the BBB-related genes, the authors should show the rationale for the selected genes. The authors focused on DEGs but do not reflect where that family of genes belongs. For example, on line 104 are the tgfb2, Marveld2, and Lrp8 related to which path or structure on the cell? 

We added gene networks for the differentially expressed genes in OXYS rats compared to Wistar rats and we pointed out pathways, significantly enriched by these genes.

o   What is the relationship of all the selected genes on section 2.1. with the BBB? The authors need to be clearer as to why they are mentioning these genes. Supporting figures/tables on the main manuscript would be very beneficial. 

BBB is enriched by these transcripts. We tried to make the grounding of the choice of genes more clear.

-       In section 2.2.: 

o   Here similar suggestion as in the previous section, the supplementary tables should be added to the main manuscript. To facilitate the reader and the overall understanding of the work. 

We tried to make the data more well-presented.

o   The authors are describing the genes differentially expressed that are observed in the supplementary tables without reflecting on what that means. Which pathway are they related to?  The authors should consider to improve this section and the previous section to make it more clear and organized for the reader. 

Thank you for the valuable comment. We added the schemes of pathways to the main text.

-       In section 2.4.: 

o   The authors have been correctly addressing the genes names using lowercase letters. However, in this section, the authors describe the animal genes using uppercase letters. The authors should uniformize all the manuscript to address the animal genes with lowercase letters. 

Thank you for the comment, we clarified this issue.

-       In section 2.5.: 

o   Line 283: the authors mentioned the mitochondria content from OXYS vs Wistar rats, and the data was not shown. However, in the discussion, the mitochondria content is referred to and discussed. Therefore, since the authors have the data, the manuscript, and the readers would benefit more if that data is shared and explained. 

Thank you for the valuable comment; we added the plot for mitochondrion content to the figure 8.

Discussion: 

-       The authors should significantly improve this section to reflect more on what is observed in the results. An example, there wasn’t any reference to the figures, to make it easier to understand the rationale. All the DEGs observed only a few of them were mentioned/discussed in this section. The authors should consider improving the explanation of the observed results. 

Thank you for the comment, we tried to make the Discussion section deeper and more clear.

-       Can the authors explain or hypothesize why the gene expression changes on OXYS rats disappeared at PND10?

The gene expression differences between OXYS and Wistar rats became less pronounced in the frontal cortex  at PND10, but became even more pronounced in the hippocampus at PND10. We tried to pay more attention to the possible explanation of age-related changes in the brain areas of OXYS rats compared to Wistar rats.

Reviewer 2 Report

In this manuscript, the authors present their results on a potential early sign of the Alzheimer’s disease (AD). The manuscript interrogates the early postnatal vascular dysfunction and its relationship to the development of AD. Scientifically, the conclusions are well-justified by appropriate experimental techniques, as well as experimental design. A few minor statistical analysis concerns need to be clarified. Language-wise, the manuscript is well-written. Every section is clear and sound. In summary, this manuscript meets the level and requirements of IJMS in all aspects. I recommend the publication after a few minor points being addressed.

11. In the Introduction section, I understand the authors want to directly address the postnatal signs of AD. However, IJMS has a broader audience than this aspect. I would strongly recommend the authors discuss the signs at different stages (middle and late) to begin with, before talking about the early-stage factors. This manuscript deserves a broader audience.

22. Still in the Introduction section, related to the first point, the authors have quite a few self-citations. The cites are appropriate but I would recommend more reference from outside the authors’ circle. For example, Line 32-34. There are tons of AD postnatal study, but the authors only cite a work in 2006, one in 2023 and a review paper. I’d rather see more detailed discussion on this topic, including who found what factor was dominant and how’s this work agrees or disagrees with it.

33. Also in the Introduction section, the authors have a series of published work on the same area. Please give a summary of the findings, as well as relating to the researches outside your group. The quality of the manuscript would be greatly improved if the authors address those above issues.

44. Line 83, “almost half” should not appear in a scientific paper. Also, please validate or clarify the significance of the afterwards results from this “almost half” changes in genes, including the meaning of those changes. How would anyone trust the conclusion with this less than 50% change in genes? This makes the results afterwards not convincing.

55. Figure 3a-c: when the authors used average values for analysis, most of the data in the 3 figures have over 20% variations. This is not appropriate and makes the analysis not convincing. Please either explaining the out-liars (e.g., especially in Figure 3c) or do an error propagation to support the analysis.

66. The original images for Blots/Gels file is awkward. Please either including it and discussing it in main text or just removing them.

Author Response

First of all, we are very grateful to the reviewers for their valuable comments and recommendations. We tried to consider every comment to make the manuscript better.

  1. In the Introduction section, I understand the authors want to directly address the postnatal signs of AD. However, IJMS has a broader audience than this aspect. I would strongly recommend the authors discuss the signs at different stages (middle and late) to begin with, before talking about the early-stage factors. This manuscript deserves a broader audience.

Thank you for the valuable comment, we added the lacking information.

  1. Still in the Introduction section, related to the first point, the authors have quite a few self-citations. The cites are appropriate but I would recommend more reference from outside the authors’ circle. For example, Line 32-34. There are tons of AD postnatal study, but the authors only cite a work in 2006, one in 2023 and a review paper. I’d rather see more detailed discussion on this topic, including who found what factor was dominant and how’s this work agrees or disagrees with it.

Thank you, we improved the discussion.

  1. Also in the Introduction section, the authors have a series of published work on the same area. Please give a summary of the findings, as well as relating to the researches outside your group. The quality of the manuscript would be greatly improved if the authors address those above issues.

We appreciate the comment. However, because OXYS rat strain was developed in our institute and is still kept in the vivarium of the institute, we characterized this strain and examined the features of OXYS rat early development.

  1. Line 83, “almost half” should not appear in a scientific paper. Also, please validate or clarify the significance of the afterwards results from this “almost half” changes in genes, including the meaning of those changes. How would anyone trust the conclusion with this less than 50% change in genes? This makes the results afterwards not convincing.

We apologize for the unclear data presentation. “Almost half” means that there are 412 BBB-enriched transcripts according to the work of Daneman and colleagues; and 200 genes belonging to this group of 412 transcripts significantly changed their expression in the frontal cortex of OXYS rats from PND3 to PND10. We tried to present the results more clearly.

  1. Figure 3a-c: when the authors used average values for analysis, most of the data in the 3 figures have over 20% variations. This is not appropriate and makes the analysis not convincing. Please either explaining the out-liars (e.g., especially in Figure 3c) or do an error propagation to support the analysis.

Thank you for the valuable comment; we added the exclusion criteria to Statistics section and corrected Figure 3.

  1. The original images for Blots/Gels file is awkward. Please either including it and discussing it in main text or just removing them.

The original images are required by the Journal. They shouldn’t be present in the main text.

Round 2

Reviewer 1 Report

The authors have significantly improved the manuscript and have addressed all my comments/suggestions.